# Holocene erosion triggered by climate change on the central Loess Plateau of China

Gang Liu<sup>1, 2</sup>, Puling Liu<sup>1,2</sup>, Hai Xiao<sup>1</sup>, Fenli Zheng<sup>1,2</sup>, Jiaqiong Zhang<sup>1,2</sup>, Feinan Hu<sup>1,2</sup>

<sup>1</sup>State Key Laboratory of Soil Erosion and Dryland Farming on the Loess Plateau, Institute of Soil and Water Conservation,
5 Northwest A&F University, Yangling 712100, China

<sup>2</sup>Institute of Soil and Water Conservation of Chinese Academy of Sciences and Ministry of Water Resources, Yangling 712100, China

Correspondence to: Gang Liu (gliu@foxmail.com)

Abstract. Understanding changes in Holocene erosion is essential for predicting soil erosion in the future. However, the quantitative response of natural erosion to Holocene climate change is limited for the Loess Plateau of China. In this study, two soil profiles were investigated in Luochuan and Yanchang sites on the central Loess Plateau of China, and four climate indicators, i.e. magnetic susceptibility, calcium carbonate content, total organic carbon content, and clay content (<0.005 mm) were analysed to describe climate change. The fitted equations using modern pedogenic susceptibility, precipitation, and temperature were used to quantitatively reconstruct paleoprecipitation and paleotemperature in the Holocene. The current

- relationship between soil erosion intensity and precipitation was determined and used to estimate historical erosion. Results indicated that the climate was coldest and driest between 12000 and 8500 cal. yr BP, then became warmer and wetter during 8500 to 5500 cal. yr BP. The warmest and wettest climate was from 5500 to 3000 cal. yr BP and was getting colder and dryer over the last 3000 cal. yr BP. Holocene erosion intensity changed with fluctuation of mean annual precipitation, and these changes were similar in both sites. However, the peak erosion values were 20790 t km<sup>-2</sup> yr<sup>-1</sup> in 7500 cal. yr BP and
- 21552 t km<sup>-2</sup> yr<sup>-1</sup> in 3300 cal. yr BP in Luochuan and Yanchang sites, respectively. Furthermore, more rapidly increasing and more severe soil erosion was predicted in Yanchang site than Luochuan with a range between 4090 and 15025 t km<sup>-2</sup> yr<sup>-1</sup> during the last 1800 cal. yr BP. This study proposed a new quantitative method to research historical soil erosion triggered by climate change, which can not only derive detailed soil erosion intensity change with variation of climate, but also provide a way to compare different areas.

## 25 1 Introduction

Soil is an important natural resource that humans rely on and civilization is based upon. The erosion of topsoil not only affects local agricultural and industrial productivity, but also serious offsite environmental problems (Palaz ón et al., 2014; Erkossa et al., 2015; Shi et al., 2016). Soil erosion is usually determined by both natural conditions, e.g. rainfall, gradient, surface cover, and soil type, and anthropogenic activities, e.g. farming, grazing, and constructing (Gabarr ón-Galeote et al.,

2013; Dai et a., 2015; Rodrigo Comino et al., 2015; Sarah and Zonana, 2015). The Loess Plateau of China, located within

the middle reaches of the Yellow River, is in the semiarid zone where natural conditions are highly susceptible to erosion (Douglas, 1989), and human activities have increased during the Holocene (Ren and Zhu, 1994; Shi et al., 2002). Owing to a combination of natural and human-induced erosion (Yu et al., 2016), it became one of the most serious soil erosion areas in the world (Fu and Gulinck, 1994). Therefore, an elementary objective of erosion control and soil conservation on the Loess Plateau should be to reduce total erosion to close to, or even lower than, the natural erosion rate. However, rates of the

- natural erosion, which are mainly determined by the geological environment and climate (Zhang et al., 2001), are not constant and are very difficult to predict (Zhao et al., 2013). During the Holocene, considering the relative stability of the geological environment on the Loess Plateau of China, climate change played a dominant role on natural erosion (Shi et al., 2002; He et al., 2006). Therefore, for assessment and prediction of the natural erosion rates in this period, it is very important
- to figure out its response to climate change.

The loess profile contains the most abundant information about the geologic evolution during the Quaternary period. It records the progression of the paleoclimate, neotectonism, paleogeography, and other important geological events in the Quaternary period in Mainland China. This profile also records the integrated processes during the evolution of global paleoclimate and paleoenvironment (Liu, 1985; An, et al., 1990). Therefore, the Loess Plateau is one of the best geological

- paleoclimate and paleoenvironment (Liu, 1985; An, et al., 1990). Therefore, the Loess Plateau is one of the best geological information carriers for global change research because it provides precious and valuable conditions in spatial and temporal dimensions (Liu et al., 1986). Several researchers have conducted such studies on the Loess Plateau of China (Maher et al., 1994; Porter et al., 2001). They have investigated the evolution of paleoclimate and paleoenvironment to provide scientific basis for forecasting future climate evolution. Some quantitative methods were also developed to estimate historical soil
- erosion on the Loess Plateau of China. One method used was to calculate historical soil erosion intensity based on the speculation of gully volume (Bai, 1994). Another common method was to compute the soil loss from the Loess Plateau of China according to the sediment in the Yellow River delta, continental shelf of the Bohai Sea, and river terrace (Ren and Zhu, 1994; Shi et al., 2002). These methods were useful but can hardly provide information on the response of natural erosion to Holocene climate change.

Numerous studies (Kirkby and Cox, 1995; Istanbulluoglu and Bras, 2006; Collins and Bras, 2008) have shown that mean annual sediment yield is a function of mean annual precipitation in various areas. Although these functions varied, they showed similar changing patterns of the relationships between sediment yield and precipitation which were called the Langbein-Schumm curve (Langbein and Schumm, 1958). These curves were primarily a function of climatic condition and

30 land use (Collins and Bras, 2008). Although other factors, e.g. soil and topography, were also crucial in determining the absolute magnitude of sediment yield from drainage basins, they mainly affected the scatter of individual points around the curve. Without regard to the human activities, Xu (2005) found that the land use factor was mainly determined by natural vegetation which was expressed by the index of net primary productivity (NPP, t ha<sup>-1</sup> yr<sup>-1</sup>). In addition, the sediment

5

delivery ratio on the Loess Plateau approaches 1 (Wei et al., 2006). Therefore, the Langbein-Schumm curve provides a possible way to evaluate past soil erosion rate if the mean annual precipitation and NPP are known.

The aims of this paper were to: (1) investigate the climate change on the central Loess Plateau of China during the Holocene; (2) reconstruct and assess paleoprecipitation and paleotemperature; (3) evaluate the response of soil erosion to Holocene climate change.

# 2 Material and Methods

# 2.1 Study area and field work