# Peer review of "Holocene erosion triggered by climate change on the central Loess Plateau of China"

_Solid Earth, 2016_

## Referee Comment (RC1) · Anonymous Referee #1 · 14 Nov 2016

This paper studies a relevant topic in the Loess Plateau of China: the variability of soil erosion during the last few thousand years, particularly in relation to changes in precipitation. The methods used are novel and the results provide good information on spatial and temporal variability of soil erosion. Nevertheless, there are some problems because the authors consider only two soil profiles as representative of a large study area. The authors do not explain which are the geomorphic factors that caused soil accumulation: Wind? Partially wind and overland flow? Mainly overland flow? Some kind of mass movement? The prevailing sediment transport and deposition process conditions very much the results, and this should be explained by the authors. The authors also need to supply information on the material that has been dated with 14C: have they directly used the soil carbon content? Concentrated pollen? -The authors dominate very well the laboratory techniques used, although some geomorphological

perspective would be interesting. -It is not clear the reason why the authors selected a 2 m profile and no other depth. -Why did you consider stationary the relationships between precipitation and erosion rates? - At the beginning of page 6: please, write "implemented".

---

## Referee Comment (RC2) · Anonymous Referee #2 · 19 Dec 2016

The manuscript by Liu et al. is puzzling. It presents a good problem: how erosion on the Loess Plateau, a prodigious sediment sink and source, is influenced by climate. It takes a logical approach by applying a Langbein-Schumm curve to tackle the erosion-climate relationship. However, once the erosion histories are available it fails to properly interpret them. As a result the paper is not publishable as it stands. Here are a few suggestions to improve it:

1. The authors should really reconsider where they send this paper for publication. EGU has a dedicated soil journal, a dedicated climate journal as well as a dedicated surface processes journal. At either one more suitable reviewers will provide more thorough reviews. I fail to see how this manuscript fits a Solid Earth journal.

2. Where are these soil profiles taken from? Are they from locations undisturbed by

human activities like ploughing, herding, deforestation, etc.? The age models look pretty good suggesting that there is no homogenization by ploughing at the top and minimal erosion, but the location and process of selecting these profiles should be described.

3. the database used to reconstruct temperature, precipitation and erosion is not openly available. For T and P the paper the authors cite a paper that does not have a proper database either. For the erosion data the authors cite Chinese reports that are not openly available for scrutiny. I wonder if the policy of the journal allows publication of a paper where such basic conditions are not fulfilled. I suggest that authors include a table showing data that lead to figure 5 providing for each point the location (lat, long), local name of watershed, primary data (magnetic susceptibility, 137CS, etc.), derived data (T, P, etc.)

4. the writing is relatively OK until the Discussion section where the text becomes very hard to read. I suggest the authors use an editorial service to correct the many mistakes peppering the text.

5. not only the language breaks down in discussion but also the analysis of data. If I look at the data presented in Fig. 4 I see that erosion was stronger during stronger aridity at one site, which contrasts with other site where erosion is minimal during the most arid interval. This does not correspond at all with the extremely brief contradictory statements of the authors: "the estimated erosion intensity during the Holocene can show a principal trend of erosion caused by precipitation" (in Discussion) and "Holocene erosion intensity changed with fluctuation of mean annual precipitation, and these changes were similar in both sites". The paper thus fails to interpret their results in my opinion providing no lesson to learn for the reader although it would be very interesting to learn why the two sites behave differently (not similarly as the authors claim).

---

## Author Comment (AC1) · 24 Dec 2016

Dear editor and reviewers,

We would like to thank you for the many useful comments related to our manuscript ID se-2016-142 entitled "Holocene erosion triggered by climate change on the central Loess Plateau of China". We have carefully considered all the points and have revised our manuscript accordingly. A more complete explanation of the changes in response to the suggestions of the reviewer is listed below. We hope that these changes now make the manuscript publishable. If you have any additional comments, we are willing to consider revision in the further.

For your convenience, in responding to the comments we have first repeated the comment denoted and then made our response.

1. This paper studies a relevant topic in the Loess Plateau of China: the variability of soil erosion during the last few thousand years, particularly in relation to changes in precipitation. The methods used are novel and the results provide good information on spatial and temporal variability of soil erosion. Nevertheless, there are some problems because the authors consider only two soil profiles as representative of a large study area. The authors do not explain which are the geomorphic factors that caused soil accumulation: Wind? Partially wind and overland flow? Mainly overland flow? Some kind of mass movement? The prevailing sediment transport and deposition process conditions very much the results, and this should be explained by the authors.

Response: Thanks for your valuable suggestion. Although the Loess Plateau of China was a large area, this study focused on its central part, where the soil accumulation process was similar and the erosion rate was different. Moreover, aeolian dust accumulation and soil erosion by overland flow, being the prevailing surface processes, have been forming the landscape of the Chinese Loess Plateau for the past 2.6 million years. However, the soil accumulation of the two research sites was seldom affected by mass movement, because the landform of two sites belongs to the tableland.

2. The authors also need to supply information on the material that has been dated with 14C: have they directly used the soil carbon content? Concentrated pollen?

Response: The 14C dates was obtained in the humin fraction which should be considered as the minimum age of the SOM. This had been explained in the manuscript (page 4, line 13-15).

3. The authors dominate very well the laboratory techniques used, although some geomorphological perspective would be interesting. It is not clear the reason why the authors selected a 2 m profile and no other depth.

Response: This manuscript focused on the time range of the Holocene. According to the results of other research (Ding et al., 1999; Zhou et al., 1994) the time range of 2 m profile in study area is about 10 ka. So, a 2-m profile was selected.

References: Ding ZL, Sun JM, Rutter NW, Rokosh D, Liu TS. 1999. Changes in sand content of loess deposits along a north-south transect of the Chinese Loess Plateau and the implications for desert variations. Quaternary Research 52, 56-62. Zhou WJ, An ZS, Head MJ. 1994. Stratigraphic division of Holocene loess in China. Radiocarbon 36, 37-46.

4. Why did you consider stationary the relationships between precipitation and erosion rates?

Response: Obviously, the relationship between erosion and precipitation was not stationary, and it was affected by soil, topography, vegetation, and human activity. However, it is hard to get the historical data of all these factors. Even though the unstationary relationship between precipitation and erosion rates was established, it is hardly to be used to predict erosion rates without the correct input data of other factors. In this study, the only way to solve this problem was to use statistical and stationary relationships.

5. At the beginning of page 6: please, write "implemented".

Response: It was revised according.

---

## Author Comment (AC2) · 24 Dec 2016

Dear editor and reviewers,

We would like to thank you for the many useful comments related to our manuscript ID se-2016-142 entitled "Holocene erosion triggered by climate change on the central Loess Plateau of China". We have carefully considered all the points and have revised our manuscript accordingly. A more complete explanation of the changes in response to the suggestions of the reviewer is listed below. We hope that these changes now make the manuscript publishable. If you have any additional comments, we are willing to consider revision in the further.

For your convenience, in responding to the comments we have first repeated the comment denoted and then made our response.

1. The manuscript by Liu et al. is puzzling. It presents a good problem: how erosion on the Loess Plateau, a prodigious sediment sink and source, is influenced by climate. It takes a logical approach by applying a Langbein-Schumm curve to tackle the erosion-climate relationship. However, once the erosion histories are available it fails to properly interpret them. As a result, the paper is not publishable as it stands. Here are a few suggestions to improve it:

Response: Thanks for the suggestion. It is very helpful to improve our paper.

2. The authors should really reconsider where they send this paper for publication. EGU has a dedicated soil journal, a dedicated climate journal as well as a dedicated surface processes journal. At either one more suitable reviewers will provide more thorough reviews. I fail to see how this manuscript fits a Solid Earth journal.

Response: Solid Earth (SE) is an international scientific journal dedicated to the publication and discussion of multidisciplinary research on the composition, structure, and dynamics of the Earth from the surface (including soil) to the deep interior at all spatial and temporal scales. This journal was recommended by Professor Artemi Cerdà, who was a topical editor of the Soil System Sciences section.

3. Where are these soil profiles taken from? Are they from locations undisturbed by human activities like ploughing, herding, deforestation, etc.? The age models look pretty good suggesting that there is no homogenization by ploughing at the top and minimal erosion, but the location and process of selecting these profiles should be described.

Response: The two soil profiles were taken from tableland. The detailed location has been described in our manuscript. The dark loessial soil is a soil type unique in the classification system of soils in China. It developed on Malan loess parent material on the Loess Plateau under semi-arid, semi-humid climate conditions, beneath grassland or forest-steppe vegetation. After pedogenic processes occurred over a long period of time, this zonal soil became one of the main arable soils in the Loess Plateau region

(Zhu et al. 1983). Due to severe soil erosion, dark loessial soils have a limited distribution over the central Loess Plateau, and currently exist only on the slightly eroded plateau surface, as well as at the top of ridges and slopes, flat areas in the gully heads, and tablelands (Zhu et al. 1983). The two studied profiles had been preserved intact because the sampling area was located on tableland where erosion is slight due to low slope gradient. However, the deep gully surrounding the tableland showed historically severe gully erosion (Fig.1). That's the reason we choose the location of soil profiles. All these have been added into our paper.

4. The database used to reconstruct temperature, precipitation and erosion is not openly available. For T and P, the paper the authors cite a paper that does not have a proper database either. For the erosion data, the authors cite Chinese reports that are not openly available for scrutiny. I wonder if the policy of the journal allows publication of a paper where such basic conditions are not fulfilled. I suggest that authors include a table showing data that lead to figure 5 providing for each point the location (lat, long), local name of watershed, primary data (magnetic susceptibility, 137CS, etc.), derived data (T, P, etc.)

Response: Yes, we can provide such a table with location and watershed name. However, the derived data was also mean annual precipitation and erosion intensity. No other primary data, e.g. magnetic susceptibility, 137Cs, was available. Thus, this table would repeat the data in Figure 5. If the journal allows, we would like to provide such a table as supporting information.

5. The writing is relatively OK until the Discussion section where the text becomes very hard to read. I suggest the authors use an editorial service to correct the many mistakes peppering the text.

Response: Thanks for your suggestion. We asked a soil scientist who was a native English speaker to revise the manuscript.

6. Not only the language breaks down in discussion but also the analysis of data. If

I look at the data presented in Fig. 4 I see that erosion was stronger during stronger aridity at one site, which contrasts with other site where erosion is minimal during the most arid interval. This does not correspond at all with the extremely brief contradictory statements of the authors: "the estimated erosion intensity during the Holocene can show a principal trend of erosion caused by precipitation" (in Discussion) and "Holocene erosion intensity changed with fluctuation of mean annual precipitation, and these changes were similar in both sites". The paper thus fails to interpret their results in my opinion providing no lesson to learn for the reader although it would be very interesting to learn why the two sites behave differently (not similarly as the authors claim).

Response: Sorry for the unclear description. It is true that the peak erosion value did not appear at the same time in two sites. That's due to the different precipitation lead to various vegetation coverage, thus a disparate soil erosion rate. The conclusion of that the estimated erosion intensity during the Holocene can show a principal trend of erosion caused by precipitation is true. However, the word "similar" in sentence that "Holocene erosion intensity changed with fluctuation of mean annual precipitation, and these changes were similar in both sites" in abstract should be changed to the word "different".